# Emergent non-Hermitian localization phenomena in the synthetic space of zero-dimensional bosonic systems

Ievgen I. Arkhipov[1, *] and Fabrizio Minganti[2, 3, †]

[1]*Joint Laboratory of Optics of Palacký University and Institute of Physics of CAS,
Faculty of Science, Palacký University, 17. listopadu 12, 771 46 Olomouc, Czech Republic*
[2]*Institute of Physics, Ecole Polytechnique Fédérale de Lausanne (EPFL), CH-1015 Lausanne, Switzerland*
[3]*Center for Quantum Science and Engineering, Ecole Polytechnique
Fédérale de Lausanne (EPFL), CH-1015 Lausanne, Switzerland*
(Dated: January 10, 2023)

Phase transitions in non-Hermitian systems are at the focus of cutting edge theoretical and experimental research. On the one hand, parity-time- ($\mathcal{PT}$-) and anti-$\mathcal{PT}$-symmetric physics have gained ever-growing interest, due to the existence of non-Hermitian spectral singularities called exceptional points (EPs). On the other, topological and localization transitions in non-Hermitian systems reveal new phenomena, e.g., the non-Hermitian skin effect and the absence of conventional bulk-boundary correspondence. The great majority of previous studies exclusively focus on non-Hermitian Hamiltonians, whose realization requires an *a priori* fine-tuned extended lattices to exhibit topological and localization transition phenomena. In this work, we show how the non-Hermitian localization phenomena can naturally emerge in the synthetic field moments space of zero-dimensional bosonic systems, e.g., in anti-$\mathcal{PT}$ and $\mathcal{PT}$-symmetric quantum dimers. This offers an opportunity to simulate localization transitions in low-dimensional systems, without the need to construct complex arrays of, e.g., coupled cavities or waveguides. Indeed, the field moment equations of motion can describe an equivalent (quasi-)particle moving in a one-dimensional (1D) synthetic lattice. This synthetic field moments space can exhibit a nontrivial localization phenomena, such as non-Hermitian skin effect, induced by the presence of highly-degenerate EPs. We demonstrate our findings on the example of an anti-$\mathcal{PT}$-symmetric two-mode system, whose higher-order field moments eigenspace is emulated by a synthetic 1D non-Hermitian Hamiltonian having a Sylvester matrix shape. Our results can be directly verified in state-of-the-art optical setups, such as superconducting circuits and toroidal resonators, by measuring photon moments or correlation functions.

## I. INTRODUCTION

Open quantum systems are ubiquitous in nature, due to the unavoidable interaction of any quantum systems with its environment. When the environment is Markovian, i.e., memory-less, a weak system-environment interaction leads to decoherence and dissipation, resulting in an irreversible dynamics where particles, energy, and information are exchanged incoherently. Within this quantum description, the concept of a real-valued energy loses its meaning because the "energy" spectrum of a dissipative quantum system becomes complex. In a striking contrast with conventional Hermitian quantum mechanics [1], the generator of the dynamics is not a Hermitian Hamiltonian operator, but rather a non-Hermitian (super)operator. Non-Hermiticity can have a number of interesting and nontrivial features [2–4], including the existence of spectral singularities called exceptional points (EPs), and their generalization to higher dimensional manifolds called exceptional lines or surfaces. At an EP, both the eigenvalues and eigenvectors of a non-Hermitian operator coalesce, a fact impossible for Hermitian operators. Even though this kind of singularity was well-known among mathematicians [2], its rising popularity

among physicists started mainly from the discovery of the pure real spectrum of parity-time ($\mathcal{PT}$) symmetric operators, leading to the characterization of $\mathcal{PT}$ non-Hermitian Hamiltonians (NHHs) [5] and to the study of phase transitions in finite-dimensional systems [5, 6]. Simultaneously, a number of experiments confirmed and demonstrated the unique properties of EPs and their influence on system dynamics [7–23] (for extensive reviews see, e.g., Refs. [3, 24, 25]).

A major obstacle to the observation of *quantum* phenomena related to non-Hermitian Hamiltonians and $\mathcal{PT}$ symmetry breaking resides in the commutation relations of quantum operators. At the quantum level, it is necessary to include dissipation (Langevin noise) in the dynamics of a non-Hermitian Hamiltonian and, de facto, breaking the $\mathcal{PT}$ symmetry [26–28]. To circumvent such a problem, two experimental strategies have been recently realized: post-selection [29] and dilation of a non-Hermitian Hamiltonian in a larger Hermitian space [30, 31]. These experiments, however, considered a single qubit, and these strategies pose significant challenges to the method scalability. In this regard, following the construction of a synthetic moment space developed in Ref. [73], *we demonstrate emergent critical behaviours in a non-Hermitian quantum simulator*. That is, we use the moments of a low-dimensional Lindbladian system capable to reproduce the dynamics of non-Hermitian lattices. This allow investigating the physics

---

* ievgen.arkhipov@upol.cz
† fabrizio.minganti@gmail.com

of non-Hermitian high-dimensional systems, in particular the non-Hermitian skin effect, through the prism of a zero-dimensional open quantum system, whose parameters can be easily tuned.

### A. Topological effects in non-Hermitian lattices

Non-Hermitian systems are known for exhibiting nontrivial topological and localization properties in condensed matter physics, particularly in 1D and higher-dimensional lattice architectures [3, 32–36]. For instance, open and periodic boundary conditions play a fundamental role in the determination of the eigenspectra of NHHs, invalidating the conventional bulk-boundary correspondence. As such, one of the main pillars of Hermitian topological physics fails to establish a quantitative correspondence between topological invariants of the bulk and the number of edge modes in a non-Hermitian system [37]. Moreover, bulk-boundary correspondence failure is intrinsically related to the so-called non-Hermitian skin effect where, in a system of size $N$, $O(N)$ edge modes, exponentially localized at the boundaries, emerge [38, 39]. Nonetheless, attempts to generalize the bulk-boundary correspondence to the case of NHHs have also been undertaken [32, 38, 40–43], provided that (possibly semi-infinite) periodic boundary conditions can be imposed. These theoretical findings have been experimentally validated in photonic platforms [44–46] (see also Refs. [36, 47] for a review). Similarly, a generalized bulk-boundary correspondence to non-periodic non-Hermitian systems with open-boundary condition has been predicted [42, 48, 49].

Several theoretical findings point at the existence of localization transitions in non-Hermitian systems, e.g., Anderson's localization in Hatano-Nelson model [50], and non-Hermitian generalizations of the Aubry-André model [51–54]. In these cases, localization is induced and controlled by uncorrelated [50] or correlated [52] disorder. Interestingly, the Anderson's localization transition has also been predicted in $\mathcal{PT}$-symmetric version of the Aubry-André model [55], revealing an interesting interplay between $\mathcal{PT}$-symmetry breaking and localization. Moreover, some hints at the emergence of rich physical phenomena in non-Hermitian systems with high-order EPs has been pointed out in Ref. [39], although associated with a Bloch formalism. The role of highly degenerate EPs on topological properties of lattice systems has also been reviewed in Ref. [36].

Some of those pioneering theoretical predictions have been experimentally confirmed, e.g., by realizing Aubry-André model in synthetic dimensions of an optical quantum walker [56, 57]. The concept of synthetic dimensions in photonics has recently attracted much interest as it allows to experimentally explore lattice physics in a more abstract, but at the same time more experimentally accessible, space [58–61]. In other words, the synthetic dimension allows exploring a variety of effects that are otherwise difficult to reach in spatial or temporal domains.

### B. Lindbladian simulator in the space of field moments

The previously cited works on non-Hermitian $\mathcal{PT}$-symmetric 1D models exhibiting localization transition [39, 55] have been derived by theoretically engineering non-Hermitian Hamiltonians to show such topological properties, i.e., with an *a priori* approach. Although being motivated by their possible experimental implementations, observing these effects can be extremely challenging in practice, because they require the construction of finely-tuned $\mathcal{PT}$-symmetric arrays of coupled cavities or waveguides. That is, the number of degrees of freedom to control inevitably increases with the lattice size. For instance, the $\mathcal{PT}$-symmetry in a lattice can be easily broken by perturbations in *any* of the lattice sites. As such, this system fragility poses serious challenges for the experimental observation of non-Hermitian topological and localization effects. Moreover, an exclusive focus on effective NHHs does not allow to correctly include quantum effects arising from the interaction of open systems with environment, leading, e.g., to quantum jumps [62–64]. The nontrivial effects arising from the latter have been pointed in recent studies considering open latices [65–67].

Here, we use a genuinely open, i.e., Lindbladian, quantum simulator [68, 69], to witness the previously described localization properties and which can be easily tuned. Indeed, in this work, we take a rather *a posteriori* approach, proposing a way to *directly access the topological and localization properties of open quantum systems via experimentally viable techniques* which do not require a fine-tuning of large lattice systems, i.e., zero-dimensional systems, and by incorporating quantum fluctuation effects. Our approach is based on the construction of synthetic spaces of field moments, and on the equivalence between this evolution and that of a particle moving in a non-Hermitian lattice, as detailed in Ref. [73]. Interestingly, recent works have also shown that *Hermitian* lattice topological phenomena can be simulated in zero-dimensional systems on the example of large-sized atomic spins [61].

### C. Motivation and novelty of the work

In this article, we demonstrate how a Lindbladian quantum simulator allows to reveal emerging lattice transition phenomena, such as a non-Hermitian skin effect, in zero-dimensional bosonic systems. In particular, we consider a minimal example of a dissipative anti-$\mathcal{PT}$-symmetric bosonic dimer [73, 104]. Furthermore, we also detail how the obtained results can be extended to arbitrary open bosonic systems exhibiting spectral singularities (exceptional points, lines, surfaces, etc. [70–72]).

Beyond the interest in simulating non-Hermitian Hamiltonians, we are motivated by the following set of fundamental questions: **(i)** A *"quantum" treatment* of the environment for finite-size systems requires including

quantum jumps. Can we use the quantum properties of bosonic operators in conjunction with quantum jumps to describe open quantum systems *beyond single-particle effective non-Hermitian Hamiltonians*? **(ii)** *Can we relate spectral (and topological) properties of (low-dimensional) open quantum systems to a more "standard" formulation of many-body physics?* **(iii)** Can we thus show a *connection between spectral ($\mathcal{PT}$-symmetrical), topological, and/or localization transitions in finite-dimensional open systems?* **(iv)** *In this quantum and dissipative framework, is it possible then to retrieve any novel and/or nontrivial effects or phenomena?*

A key point, for answering all those questions, is the use of a high-dimensional *synthetic functional or operator space* (from now on, synthetic space for the sake of brevity). Within this synthetic space, and for the simplest case of a dimer, the evolution matrix ruling the dynamics of higher-order field moments of the dimer mimics a NHH of a particle in a 1D lattice [73]. Starting from the mathematical construction of Ref. [73], remarkably, we are able to connect the physics of EPs in quadratic systems to that of localization transitions, even in 0D. The higher the moments considered, the larger the corresponding 1D synthetic lattice, in analogy to operator spreading in standard condensed-matter physics [74, 75], thus allowing to **(i)** correctly describe the "localization" properties of dissipative open system using **(ii)** a standard formalism of one-particle Hamiltonians (although non-Hermitian) to describe the properties of the open quantum systems. Because of the infinite dimension of Hilbert space of bosonic fields, the lattice can be extended to any size $N$. Due to an algebraic relation between each of these moment spaces, we can **(iii)** relate the presence of a spectral phase transition to a localization transition of a (quasi)particle moving in the 1D synthetic lattice. **(iv)** This new synthetic space approach provides a new interpretation of non-Hermitian Hamiltonians [76], and allows exploring phenomena which, otherwise, could be very difficult to witness in standard non-Hermitian mechanics.

We remark that an alternative approach to create synthetic high-dimensional space based, instead, on a decomposition of Fock Hilbert space of *phenomenological* low-dimensional non-Hermitian Hamiltonian operators, have been proposed, e.g., in [30, 77, 78]. However, working exclusively in the Fock space of dissipative systems inevitably invokes a postselection procedure, when one has to discard quantum jumps in the system dynamics, whose rate rapidly increases with the system size, and which, thus, makes such an approach *practically* unfeasible for many particle systems. Moreover, in those works, the emergent localization transitions in such constructed synthetic space have not been investigated.

## D. Organization of the paper

The paper is organized as follows. In Sec. II, we introduce an anti-$\mathcal{PT}$-symmetric bosonic dimer, and demonstrate how its space of higher-order field moments can be mapped to a synthetic 1D $\mathcal{PT}$-symmetric lattice. In Sec. III, we analyze the topology, localization and emergent $\mathbb{Z}_2$ non-Hermitian skin effect in the synthetic 1D space of the dimer. In Sec. IV we detail how such localization phenomena can be directly simulated and observed by measuring photon moments or correlation function of the dimer fields. We also discuss the possible extension of the emergent localization phenomena to arbitrary zero-dimensional bosonic systems with EPs and give an outlook for future research in Sec. V. Conclusions are drawn in Sec. VI.

## II. THEORY

Under quite general hypotheses [79], the evolution of a density matrix of a open quantum system is described by a completely positive and trace-preserving (CPTP) linear map. That map, in turn, allows to derive evolution matrices for moments of both system Hermitian (observables) and non-Hermitian operators. The dynamics, which governs those operator moments, can be mapped to a certain non-Hermtitian lattice, by quantizing the corresponding moments. In other words, the synthetic space of higher-order operator moments becomes equivalent to spatial space of a one-particle described by a 1D NHH. This property stands behind the idea of our quantum simulator: we show that the dynamics of system operators of a (zero-dimensional) open quantum system can correspond to that of a particle moving in a spatial space under the action of a (1D) non-Hermitian Hamiltonian [see also Fig. 1(c)]. This is true for any given quadratic Lindbladian, as shown in Ref. [73].

As a minimal example of a Lindbladian quantum simulator, we consider an anti-$\mathcal{PT}$-symmetric bosonic dimer coupled to a Markovian environment, consisting of two incoherently coupled quantum fields represented by boson annihilation operators $\hat{a}_1$ and $\hat{a}_2$ and with frequencies $\omega - \Delta$ and $\omega + \Delta$, respectively. The following results can be easily extended to $\mathcal{PT}$-symmetric dimers as well. By tracing out the degrees of freedom of the environment and passing in the frame rotating at the frequency $\omega$, the equation of motion of the system's reduced density matrix $\hat{\rho}$, which contains all the statistical information of the dimer, has the following (Gorini-Kossakowski-Sudarshan-) Lindblad form [80–83]:

$$\frac{\mathrm{d}}{\mathrm{d}t}\hat{\rho}(t) = \frac{1}{i\hbar}\left[\hat{H}, \hat{\rho}(t)\right] \\ + \sum_{j,k=1,2} \gamma_{jk}\left(2\hat{a}_j\hat{\rho}(t)\hat{a}_k^\dagger - \hat{a}_k^\dagger\hat{a}_j\rho(t) - \rho(t)\hat{a}_k^\dagger\hat{a}_j\right),$$

(1)

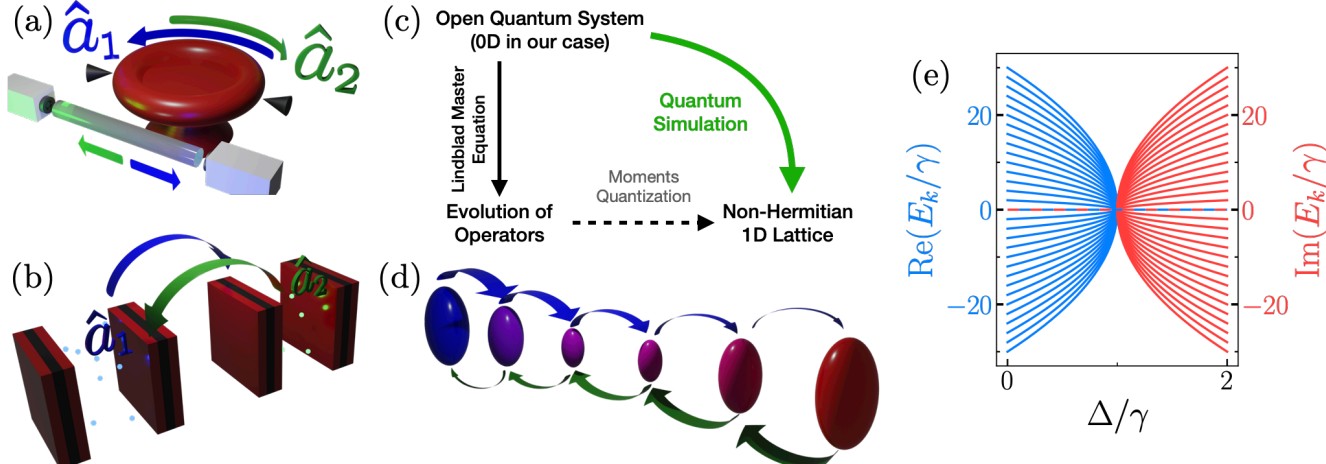

FIG. 1. Possible experimental setups which obey Eq. (1) and resulting dynamics. (a) In a microtoroidal resonator, by appropriately placing nanotips it is possible to introduce an asymmetric scattering between the clockwise and counter-clockwise propagation modes. (b) Similarly, in optical systems (such as cavitiy or circuit QED) cascaded systems can be engineered to have an asymmetric dissipation rate between two modes or two cavities (represented as the modes $\hat{a}_1$ and $\hat{a}_2$ in the panel). (c) A scheme of the proposed way to use the Lindbladian system as a quantum simulator, as described in the main text. (d) The dynamics of Eq. (1) corresponds to the NHH in Eq. (2), describing a particle moving in the 1D lattice. The onsite energy (presented by ellipsoids) changes in modulus (in size) and in sign [from positive (red color) to negative (blue)]. The intersite right (blue) and left (green) couplings are asymmetric and site-dependent, as signalled by the different width of the arrows. (e) Real (blue) and imaginary (red) parts of the energies $E_k$ of Eq. (2) versus $\Delta$, according to Eq. (3) for a chain with odd number of sites $n = 31$. At $\Delta = \Delta_{EP} = \gamma$, the system displays a $n$th-order EP for a $n = N + 1$ site lattice. At the EP, the energies coincide and corresponding eigenvectors coalesce (not shown here). For a lattice with odd number of sites, the spectrum has always a zero-energy mode in both real and imaginary parts (dashed red and blue line), while for an even number of sites the spectrum does not have a zero-energy eigenmode.

where $\hat{H} = \Delta(\hat{a}_1^\dagger \hat{a}_1 - \hat{a}_2^\dagger \hat{a}_2)$, $\gamma_{jk}$ are real and positive elements of a $2 \times 2$ decoherence matrix $\hat{\gamma}$, and $\hbar$ is the reduced Planck's constant. The diagonal elements of the decoherence matrix account for the inner mode dissipation, whereas off-diagonal terms are responsible for the incoherent mode couplings. Such a Lindblad master equation naturally emerges in several experimental platforms. For instance, the asymmetric intermode dissipation can be realized in cascaded optical systems [84], where asymmetric dissipation favors a unidirectional flow of particles [c.f. Fig. 1(a)], in microtoroidal resonators [24, 85], where modes incoherent scattering between clockwise and counter-clockwise modes can be engineered using nanotips [see Fig. 1(b)], parametric optical oscillators [86], and other optical setups [87–90].

For the sake of simplicity, in what follows, we assume that the decoherence matrix $\hat{\gamma}$, in Eq. (1), is symmetric, i.e., $\gamma_{12} = \gamma_{21} = \gamma$, and $\gamma_{11} = \gamma_{22} = \Gamma$. For Eq. (1) to be stable, the decoherence matrix must be positive-definite and $\det\hat{\gamma} > 0$ (i.e., $\Gamma > \gamma$). Although physical systems must respect this condition, in the mathematical treatment the positive-definiteness property can be eased and we can safely set $\Gamma = 0$ in the following. Indeed, one can make $\det\hat{\gamma} < 0$ by properly gauging away the decoherence rate $\Gamma$ [24], meaning that one starts working in the local frame which is globally decaying. Similarly, zero-net losses can be induced by balanced gain, whose quantum noise can be safely discarded when dealing with

non-Hermitian forms of the dimer field moments as done in, e.g., Ref. [73] (although too large gain leads to instabilities, and the zero-net loss assumption is valid either for short times or for nonlinear systems [15]).

The anti-$\mathcal{PT}$-symmetric nature of the system is reflected in the time dynamics of the $N$th order moments $\boldsymbol{A} = \left( \langle \hat{a}_1^N \hat{a}_2^0 \rangle, \langle \hat{a}_1^{N-1} \hat{a}_2 \rangle, \ldots, \langle \hat{a}_1^0 \hat{a}_2^N \rangle \right)^T$, which is governed by the corresponding evolution matrix $M$. One has $i\partial_t \boldsymbol{A} = iM\boldsymbol{A}$ [the $N$th order field moments are a closed space under the action of Eq. (1) and $\{iM, \mathcal{PT}\} = 0$, where curled brackets denote anticommutator (see Appendix A and Refs. [15, 73]). Note the latter condition implies that the matrix $M$ commutes with $\mathcal{PT}$ operator, i.e., it is $\mathcal{PT}$ invariant.

Remarkably, the dynamics of $\boldsymbol{A}$ can be "quantized", and the evolution matrix $M$ describes the equation of motion of one particle in a non-Hermitian 1D chain with $(N + 1)$ sites [91]. In other words, the synthetic space of the dimer higher-order field moments is equivalent to the NHH $\hat{H}_{1D}$ of a 1D chain [c.f. Fig. 1(c,d)], and reads (see Appendix A):

$$\hat{H}_{1D} = \sum_{j=0}^{N} \mu_j \hat{c}_j^\dagger \hat{c}_j + \sum \gamma_j^R \hat{c}_{j+1}^\dagger \hat{c}_j + \sum \gamma_j^L \hat{c}_j^\dagger \hat{c}_{j+1}, \quad (2)$$

where $\hat{c}_j^\dagger(\hat{c}_j)$ is the creation (annihilation) operator at site $j$, the onsite energy $\mu_j = -i(N - 2j)\Delta$, and position-

dependent right and left couplings are $\gamma_j^{\mathrm{R}} = -(N-j)\gamma$ and $\gamma_j^{\mathrm{L}} = -j\gamma$, $j = 0, \ldots, N$, respectively. This 1D NHH is invariant under parity-time transformations. In its matrix form, $\hat{H}_{1\mathrm{D}}$ in Eq. (2) is a tridiagonal Sylvester matrix [92]. Moreover, $\hat{H}_{1D}$ is also characterized by chiral symmetry. Chirality is characterized by a unitary operator $\chi$ such that $\chi \hat{H}_{1\mathrm{D}} \chi^\dagger = -\hat{H}_{1\mathrm{D}}$, and $\chi\chi^\dagger = \chi^\dagger\chi = \chi^2 = 1$ [33, 36, 93] (see Appendix A for the explicit form of $\chi$).

Note that, due to the $\mathcal{PT}$-symmetry and the absence of translational invariance of the matrix $M$, it is physically meaningless to impose periodic boundary condition in such synthetic 1D lattice. Of course, one can mathematically "glue" the two ends of the 1D chain in such a way that the $\mathcal{PT}$-symmetry is preserved, but this would amount to a complete change of the model in Eq. (1) by introducing *ad hoc* terms. Moreover, the addition of such terms has no physical basis, since the 1D chain is defined in the field moments space. As such, only open boundary conditions in Eq. (2) can be well-justified from a physical point of view.

It is also worth noting that NHH similar to that in Eq. (2) was *a priori* introduced and studied in Refs. [94], but whose construction required the presence of $n$-coupled finely tuned cavities. Here, instead, we show that such NHH can naturally arise in the moments space of system operators of a dimer and, thus, reveal its physical origin.

Importantly, when dealing with any *a priori* NHH, the knowledge of both its left and right eigenvectors becomes necessary in order to assign a physical meaning to the inner product of the corresponding Hilbert space [95]. Nevertheless, in the case we are studying, the *a posteriori* NHH $\hat{H}_{1\mathrm{D}}$ emerges only in the synthetic space of field moments, and the underlying physics is described by the well-defined Lindblad master equation, where such precautions are not needed. Indeed, the evolution matrix is defined for c-number vectors, not q-number valued eigenstates of a quantum Hamiltonian. As such, the right eigenvectors of the 1D NHH correspond to the well-defined right eigenvectors of the field moments evolution matrix. Consequently, one would need the left eigenvectors of $\hat{H}_{1\mathrm{D}}$ only for mathematical manipulations [96, 97] not employed in this article.

We are now in position to investigate in detail the spectral properties of the NHH $\hat{H}_{1\mathrm{D}}$ associated with EPs, $\mathcal{PT}$-symmetry breaking, and other topological and localization phenomena.

## III. RESULTS

### A. $\mathcal{PT}$-symmetry breaking and topological phase transition of $\hat{H}_{1\mathrm{D}}$

The eigenvalues $E_k$ of $\hat{H}_{1\mathrm{D}}$ in Eq. (2), for a chain with $n = N + 1$ sites, are [73]:

$$E_k = (N - 2k)\sqrt{\gamma^2 - \Delta^2}, \quad k = 0, \ldots, N. \qquad (3)$$

The explicit expressions for the components of the corresponding right eigenvectors $|\psi_k^{\mathrm{R}}\rangle = \left[\psi_k^{\mathrm{R}}(0), \ldots, \psi_k^{\mathrm{R}}(N)\right]^T$ are given in Eq. (B1) in Appendix B. The energy spectrum is symmetric with respect to the zero-energy $E = 0$, i.e., $E_k = -E_{N-k}$, a consequence of the chiral symmetry of the system. As it follows from the relation in Eq. (3), whenever the number of sites $n$ is odd, the system possesses a zero-energy eigenmode [see Fig. 1(e)], in a close analogy with the Hatano-Nelson model with open-boundary condition [50, 98]. Also, similar spectral behaviour of $\mathcal{PT}$-symmetric 1D lattices, but with particle-hole symmetry, has been observed in Refs. [99, 100]. However, those models are based on spatial 1D chains with sublattice structures, and deal with the multiband Bloch formalism.

According to Eq. (3), when $\Delta = \Delta_{EP} = \gamma$, the spectrum of $\hat{H}_{1\mathrm{D}}$ becomes completely degenerate, i.e., $E_k = 0$ for all $k$. Furthermore, all the eigenvectors also coalesce [c.f. Eq. (B1)], meaning that the system has an EP of $n$-th order for a 1D lattice with $n$ sites. At the EP, the NHH $\hat{H}_{1\mathrm{D}}$ undergoes a spectral phase transition, from an exact- ($\Delta < \Delta_{EP}$, the energies are real-valued) to a broken-$\mathcal{PT}$ phase ($\Delta > \Delta_{EP}$, the energies are purely imaginary).

For zero-dimensional systems, the $\mathcal{PT}$-symmetric spectral transition is accompanied by a topological transition, which can be described by a $\mathbb{Z}_2$ invariant, determined by the sign of the determinant of the NHH [32]. In the considered case, for $\mathcal{PT}$-symmetric 1D NHH in Eq. (2), the same $\mathbb{Z}_2$ invariant can be applied as well. Indeed, the $\mathbb{Z}_2$ invariant of $\hat{H}_{1\mathrm{D}}$ is inherited from an effective 0D NHH $\hat{H}_{\mathrm{eff}}$, which can be derived from the Eq. (1) (see also Appendix C). In particular, when the number of sites is even, i.e., $n = 2k$ for $k$ odd, the sign of the determinant of the NHH $\nu$ is a $\mathbb{Z}_2$ invariant, i.e., $\nu = \mathrm{sgn}[\det(\hat{H}_{1\mathrm{D}})] = \pm 1$, as it directly follows from Eq. (3). The phases with $\nu = -1$ ($\nu = 1$) correspond then to the exact (broken) $\mathcal{PT}$-symmetric phase. This topological transition occurs at the $n$-degenerate EP, where the system energy is $E = 0$ (both in imaginary and real part). We recall, that the NHH $\hat{H}_{1\mathrm{D}}$ does not possess spatial invariance, hence no periodic-boundary conditions can be imposed; thus, the Bloch-band or generalized Brillouin zone formalism cannot be used to calculate topological invariants such as winding numbers [32, 33, 38, 40, 41]. Additionally, for such a system with OBC, one could also try to utilize a real space topological invariant, e.g., a biorthogonal polar-

ization [48]. The biorthogonal polarization is calculated knowing both the left and right eigenvectors, and is applied to the zero-energy mode. However, since in our case the zero-energy mode is either absent (for an even-site chain) or it exists in the whole parameter space (for an odd-site chain), no topological features can be deduced from it [98].

## B. Exceptional-point-induced localization transition and non-Hermitian skin effect

We characterize now an EP-induced localization transition in terms of right eigenmodes of the NHH $\hat{H}_{1D}$. Since the entire physics is determined by the ratio $\Delta/\gamma$, without loss of generality in the following analysis we choose to vary $\Delta$, leaving $\gamma$ fixed. Given the chiral symmetry of the NHH, one has $|\psi_k^R(i)| = |\psi_{N-k}^R(N-i)|$, regardless whether the $\mathcal{PT}$-symmetry is broken or not [the chirality condition $E_k = -E_{N-k}$ holds true for both purely real and imaginary energies, according to Eq. (3)] (see also Appendix B for details).

In the exact $\mathcal{PT}$-phase, the chain eigenvectors can be either extended, or boundary-localized eigenmodes. In the $n \to \infty$ limit, the extended eigenmodes are characterized by a vanishingly small probability distribution over all sites of the chain, while the boundary eigenmodes tend to localize at both edges of the chain [101]. This simultaneous localization at both edges is because these eigenmodes are also eigenvectors of the $\mathcal{PT}$ operator in the exact $\mathcal{PT}$-phase, and the probability distribution of the eigenmodes is symmetric with respect to the center of the lattice, i.e, $|\psi_k^R(i)| = |\psi_k^R(N-i)|$ [see Fig. 2 and Eq. (B1)]. That is, each boundary eigenmode can also be treated as a superposition of two degenerate edge modes located at opposite sides of the lattice [94]. According to Eqs. (B1) and (B2), the most delocalized eigenmodes are $|\psi_0^R\rangle$ and $|\psi_N^R\rangle$, whose energy are $E_0$ and $E_N$, respectively. These states are uniformly distributed ($|\psi_0^R(i)| =$const) in the whole unbroken $\mathcal{PT}$ phase [see Fig. 2(a)].

In the deep exact $\mathcal{PT}$-phase, i.e., when $\Delta \ll \Delta_{EP}$, there are $O(n)$ boundary eigenmodes whose energy is $E_N < E < E_0$ [see Figs. 2(b,c)]. Notably, the closer the energy to zero, the more localized the corresponding eigenmode at both edges. The parameter $\Delta$ plays a similar role to that of the correlated disorder [55]. By increasing $\Delta$, the "tails" of the boundary modes steadily penetrate into the bulk, eventually becoming completely delocalized right below the EP [$\Delta \lesssim \Delta_{EP}$, c.f. Figs. 2(b,c)]. Since the EP is of the $n$th order, this implies that all $N + 1 = n$ eigenemodes collapse to a single and completely delocalized eigenvector $|\psi_{EP}^R\rangle = |\psi_0^R\rangle = |\psi_N^R\rangle$. In other words, this is a localization crossover induced by the EP spectral singularity, inducing a complete eigenmode overlap right before the EP.

In the broken $\mathcal{PT}$ phase, right above the EP ($\Delta \gtrsim \Delta_{EP}$), a number $O(n)$ of eigenmodes experience a sud-den localization transition due to the EP nonanaliticity (see Fig. 2). The modes whose Im $(E_k)$ is negative (positive) localize at the right (left) edge of the chain. This is a manifestation of the *non-Hermitian skin effect*, where the system eigenmodes become exponentially localized at the edges [36, 42, 50, 98, 102]. While these $O(n)$ eigenmodes localize, $O(1)$ number of modes, whose eigenenergies lie closer to zero, are delocalized but with a high-occupation at the center of the chain [see Fig. 2(c)]. Note that the skin modes appear at both edges, contrary to Hatano-Nelson and non-Hermitian Su-Schrieffer-Heeger models [50, 103]. Such an unusual and simultaneous accumulation of skin modes at the two edges has also been dubbed as $\mathbb{Z}_2$ skin effect [42]. Since, above the EP, the eigemodes must also exhibit a large spatial overlap, such an overlay in the broken-$\mathcal{PT}$ phase is a spatial accumulation of the modes at the edges [36]. The fact that the eigenmodes pile up at both edges of the chain is dictated by the system chirality: $|\psi_k^R(i)| = |\psi_{N-k}^R(N-i)|$.

Finally, in the deep broken $\mathcal{PT}$-phase ($\Delta \gg \Delta_{EP}$), when the intersite couplings become irrelevant, i.e., $\Delta \gg \gamma$, the number of skin modes, localized at the edges, steadily decreases. That is, the eigenmode, with $E_k \neq 0$, detaches from right (left) edge, when $k \leq n/2$ ($k > n/2$), and starts gradually 'moving' inside the chain, with increasing $\Delta$, and then localizes at the corresponding site $N - k$ (see Fig. 2). This $\Delta$-induced localization is akin to other "disorder"-induced localization transitions [55]. One, thus, concludes that the *two edge modes*, with the largest absolute energies $|E_{0,N}|$, *remain at the edges in the whole broken-$\mathcal{PT}$-phase* independently of $\Delta$. Hence, the $\mathbb{Z}_2$ topological invariant, defined by the determinant of the 1D NHH, can be associated with the existence of these two skin edge modes in the broken-$\mathcal{PT}$ regime. Indeed, the two modes $|\psi_{0,N}^R\rangle$ stem directly from the two eigenmodes of the original zero-dimensional $\mathcal{PT}$-symmetric NHH (expressed via a $2 \times 2$ matrix), upon which the given 1D NHH $\hat{H}_{1D}$ is built (see Appendix C). Since the two eigenmodes of the 0D NHH are related to the $\mathbb{Z}_2$ topological invariant [32], so will be the two modes $|\psi_{0,N}^R\rangle$ of the 1D NHH $\hat{H}_{1D}$. We conclude that the *robustness of these two edge modes* with respect to the $\Delta$-correlated "disorder" could be an *experimental demonstration of the topological nature* of the $\mathcal{PT}$ symmetry breaking described here.

The robustness of these two modes to the perturbations can be further explained as following. Since $|\psi_{0,N}^R\rangle$ are derived from the two modes of the $\mathcal{PT}$-symmetric 0D NHH, this means that any perturbations induced in the original 0D NHH, and which preserves the $\mathcal{PT}$-symmetry, would not affect the eigenmodes $|\psi_{0,N}^R\rangle$ of the 1D NHH, i.e., they stay immune to the emerging perturbations in the 1D NHH, which originate from the 0D Hamiltonian.

In the case of an odd-site chain, the zero-energy mode exist in the whole system parameter space, and, thus, fails to reveal any topological features [98]. However, the peculiarity of a zero-energy eigenmode $|\psi_{N/2}^R\rangle$ with $E = 0$ consists in the fact that it is the only one which

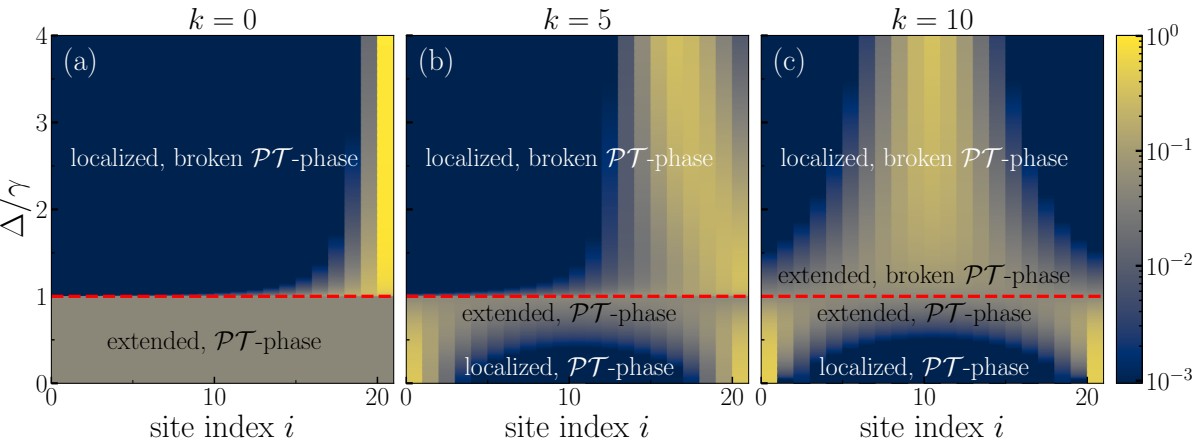

FIG. 2. Intensity distribution of the components of right eigenvectors $|\psi_k^{\mathrm{R}}(i)|^2$ with (a) $k = 0$, (b) $k = 5$, and (c) $k = 10$ in a 1D $\mathcal{PT}$-symmetric lattice with $n = 21$ sites for various values of $\Delta/\gamma$, according to Eq. (B1). The eigenvector with eigenenergy $E_0$ is completely delocalized in the whole $\mathcal{PT}$-symmetric phase $\Delta/\gamma < 1$ [panel (a)]. In the broken $\mathcal{PT}$-phase, however, it is localized at the right edge. The eigenvectors with $k \neq 0, N$, on the contrary, are localized at both boundaries in the deep exact $\mathcal{PT}$-phase ($\Delta/\gamma \ll 1$), and experience delocalization right below the EP $\Delta_{\mathrm{EP}} = \gamma$ [panels (b) and (c)], because all eigenvectors condense to an extended eigenvector at the EP. Moreover, all the eigenvectors except zero-energy mode with $k = N/2$, after crossing the EP ($\Delta/\gamma > 1$), exhibit strong localization at either left or right edges, undergoing the so-called non-Hermitian skin effect [panels (a) and (b)]. Zero-energy mode, instead, remains delocalized right above the EP [panel (c)]. In the deep broken $\mathcal{PT}$-phase ($\Delta/\gamma \gg 1$), each eigenvector $\psi_k^{\mathrm{R}}$ tends to localization over the site with index $N - k$ [panels (a)-(c)]. We do not plot the eigenvectors with opposite eigenenergies $E_{N-k} = -E_k$, because they are identical to those shown apart from a symmetric reflection with respect to the center of the chain, a consequence of the chiral symmetry of the NHH. In all panels, the red dashed line denotes the $N$th order EP.

does not lose its symmetry with respect to the center [see Fig. 2(c)]. This behaviour of the zero-energy mode can also be explained by the chiral symmetry. It is the only eigenmode which is also an eigenmode of the chiral operator $\chi$, which implies $|\psi_{N/2}^{\mathrm{R}}(i)| = |\psi_{N/2}^{\mathrm{R}}(N - i)|$ (see Appendix B).

To corroborate quantitatively the observed localization transition induced by the EP, we calculate the inverse participation ratio (IPR) for the right eigenmodes [97]. The formula for the IPR simply reads as

$$
\mathrm{IPR}(E_k) \equiv \left\{ \frac{\left[ \sum_i \left| \psi_k^{\mathrm{R}}(i) \right|^2 \right]^2}{\sum_i \left| \psi_k^{\mathrm{R}}(i) \right|^4} \right\}^{-1}. \tag{4}
$$

The values of the IPR can range from $O(1/n)$, which designates the pure extended states, to $O(1)$, which corresponds to the localized states. In other words, the IPR for extended states are characterized by the limit $\lim_{n \to \infty} \mathrm{IPR}(E) = 0$. Whereas for localized states the IPR acquires finite nonzero values regardless of the lattice size $n$.

We show the results of the calculation of the IPR as a function of $\Delta/\gamma$ and the number of sites $n$ for various right eigenmodes in Fig. 3(a) and Figs. 3(b)-(c), respectively. As one can see in Fig. 3(a), the IPR clearly exhibits a second-order discontinuity at the EP $\Delta_{EP}$, and

the further away the energies from $E = 0$ the sharper the change in the IPR. This characteristic jump accounts for the localization transition, from extended states just below the EP (IPR is of order $O(1/n)$), to the localized skin modes right above the EP (IPR of order $O(1)$) [see also Figs. 3(b),(c)]. Such behavior of the IPR in the vicinity of the EP is most clearly observed for the eigenmodes $k = 0$ and $k = N$ (dark blue solid curve), whose energies lie at opposite sides of the energy spectrum, as confirmed in Fig. 2.

The zero-energy eigenmode ($k = 10$) and modes with $E \approx 0$ (e.g., $k = 8$), for which $k \approx N/2$, do not demonstrate such a prominent change in the IPR [see Fig. 3(a)]. The IPR remains mostly unchanged right above the EP, indicating that these modes stay (mostly) delocalized even after crossing the EP. The fact that those states remain delocalized is also confirmed by Fig. 3(c), where $\mathrm{IPR}(E_k \approx 0) \to 0$ when $n \to \infty$ [magenta curve in Fig. 3(c)]. All eigenmodes become localized in the deep broken $\mathcal{PT}$-phase when $\Delta/\gamma \to \infty$. On the other hand, the two *topological* edge modes with $E_{0,N}$, discussed above, remain localized in the whole broken $\mathcal{PT}$-phase, characterized by ever-growing IPR, saturating at the value $\mathrm{IPR}(E_{0,N}) = 1$ in the deep broken phase [$k = 0$ in Fig. 3(a)].

In the broken-$\mathcal{PT}$ phase, the IPR can *quantitatively* detect a "detachment" of a given skin mode from either edge and its subsequent localization inside the bulk, as

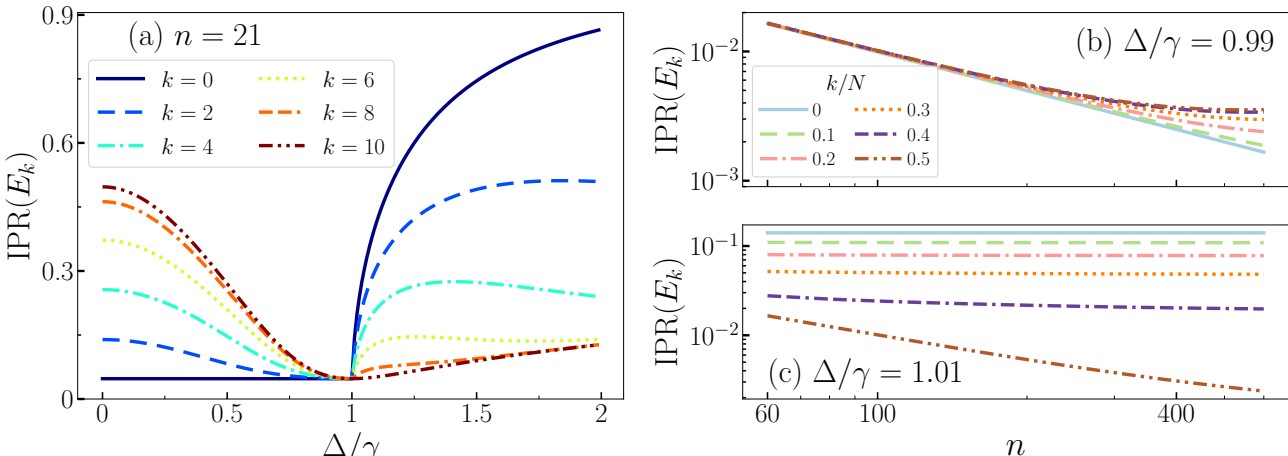

FIG. 3. IPR for various right eigenvectors with eigenenergies $E_k$ in a 1D $\mathcal{PT}$-symmetric chain. (a) IPR as a function of $\Delta/\gamma$, according to Eq. (4), for a lattice with $n = 21$ sites. Apart from the spectral and topological transition, the EP of the $n$th order also signals the localization transition, where the IPR experiences a continuous jump for delocalized states in the vicinity of the EP [e.g., blue solid, cyan dashed and green dash-dotted curves in panel (a)]. In the deep exact and broken $\mathcal{PT}$-phase, there are $O(n)$ boundary localized states, characterized by the IPR values of order $O(1)$. In the vicinity of the EP, right below and above the EP, there $n$ extended and $O(n)$ localized eigenmodes, respectively. (b)-(c): IPR as a function of the number of sites $n$ in the system: (b) right below the EP with $\Delta/\gamma = 0.99$, and (c) right above the EP with $\Delta/\gamma = 1.01$. Right below the EP [panel (b)], all $n$ eigenmodes becomes delocalized, where the IPR is decreasing with the number of sites as $O(1/n)$. Right above the EP [panel (c)], there are $O(n)$ eigenstates localized at the left or right edges of the chain for which the IPR saturates with the number of sites and attains a finite value of order $O(1)$.

was *qualitatively* described earlier. Indeed, the IPR displays a local maximum for edge-localized skin modes with $0 < |E| < |E_0|$ right above the EP, in the broken-$\mathcal{PT}$ phase [see e.g., $k = 2$, $k = 4$, and $k = 6$ in Fig. 3(a)]. The larger the energy of a skin mode, the larger the value of $\Delta/\gamma$, at which a local maximum in the IPR appears [see Fig. 3(a)]. Note that the increasing (decreasing) values of IPR signal the tendency of a given mode to its localization (delocalization). As such, the local maxima in Fig. 3(a) indicates the point where a given skin mode detaches from an edge and start its "migration" inside the bulk. Increasing $\Delta$, the state gets more and more delocalized until the eigenmode with $E_k$ stops "moving" in the chain and settles down at the corresponding site with index $N - k$. At this point, further increasing $\Delta$ amounts again to the increasing strength of a mode localization in the system, i.e., its IPR start attaining larger values. In other words, the edge-localization–detachment–migration—bulk-localization behavior can be understood as the competition between the $\mathcal{PT}$-symmetry breaking edge-localization effect and the $\Delta$-induced bulk localization, as also can be seen in Fig. 2(b).

## IV. SIMULATING NON-HERMITIAN MECHANICS WITH HIGHER-ORDER FILED MOMENTS

Having shown the presence of a localization transition and the formation of skin eigenmodes in the synthetic space of the $\hat{H}_{\text{1D}}$, here we discuss how these critical phenomena reflect onto the original anti-$\mathcal{PT}$-symmetric two-mode dimer. That is, the localization of its field moments in time evolution. In particular, we highlight how the studied phenomena can be directly observed and probed in the state-of-art experimental setups by measuring either higher-order field moments or correlation functions of light generated in the two-mode system.

As it was stressed in Sec. II, the $\mathcal{PT}$-symmetric form of the moments matrix $M$, which governs the time dynamics of higher-order field moments, can emerge either in a local reference frame, which is globally decaying with a rate $\Gamma$, or in a dimer with balanced gain and loss in both modes, which results in the zero-net dissipation. Note that in the former case, when returning to a laboratory frame, the decaying rate of the $N$th-order field moments becomes proportional to $N\Gamma$, i.e., the higher the moment the higher the dissipation (see Appendix A). Also, in the deep broken-$\mathcal{PT}$ phase, it may be necessary to include nonlinear terms to avoid nonphysical divergences in the long time limit of the system evolution [15]. In the latter case, a linear system model is valid only within a short time interval.

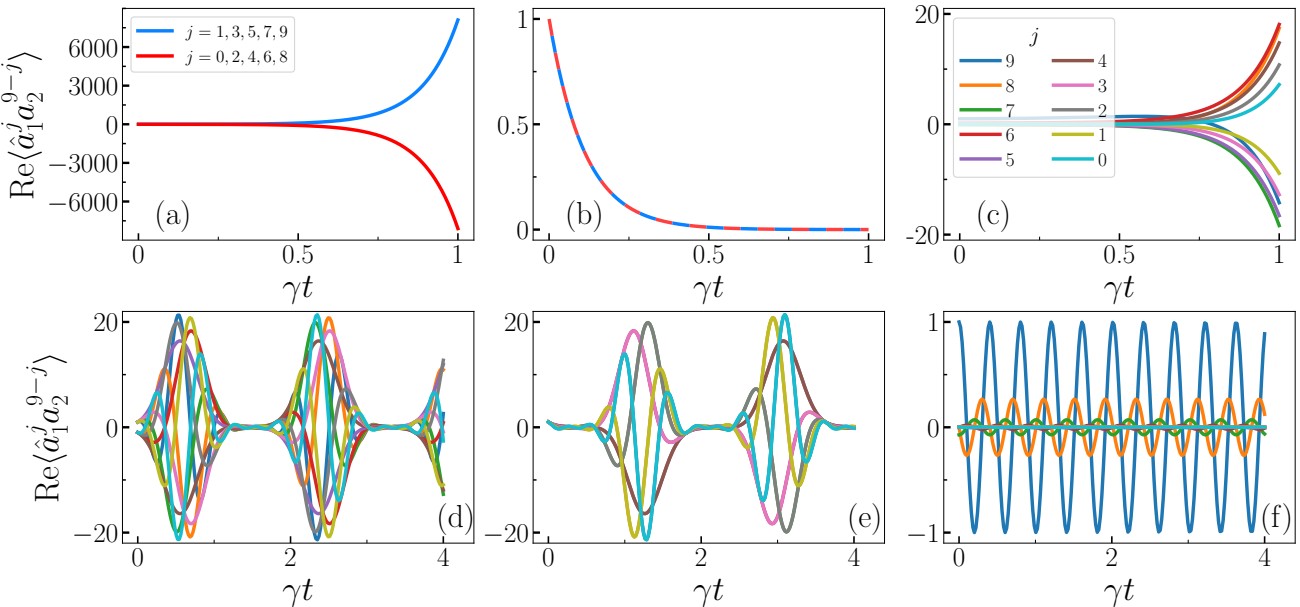

FIG. 4. Time evolution of a real part of higher-order field moments $\langle \hat{a}_1^j \hat{a}_2^{9-j} \rangle$ (up to ninth order) of the anti-$\mathcal{PT}$-symmetric two-mode bosonic system governed by the evolution matrix $M$ for the eigenvectors $|\psi_0^R\rangle$ and $|\psi_N^R\rangle$ of the $M$, given in Eq. (B2) and Eq. (A1), respectively. (a)-(c): the field moments in the broken anti-$\mathcal{PT}$-phase with $\Delta = 0$; and (d)-(f): in the exact anti-$\mathcal{PT}$-phase with $\Delta/\gamma = 2$. (a),(b) the field moments forming the extended eigenstates in the broken anti-$\mathcal{PT}$-phase and initialized in coherent states $|\alpha = 1\rangle|\beta = -1\rangle$ and $|\alpha = 1\rangle|\beta = 1\rangle$, respectively. These eigenstates either exponentially grow with the rate $|E_0|$ in (a), or decay with the same rate $|E_N| = |E_0|$ in (b). (d)-(e): the same initialized coherent states but in the exact anti-$\mathcal{PT}$-phase, where they are not the system eigenstates anymore. (f) field moments forming the edge eigenstate $|\psi_0^R\rangle$ in the exact anti-$\mathcal{PT}$-phase, initialized in the coherent state $|1\rangle|-0.268i\rangle$. The color legends in panels (d)-(f) are the same as in panel (c). For the sake of clarity, it is assumed that the dimer evolves in a local frame where $\Gamma = 0$, but which can globally decay at some rate $\Gamma \neq 0$ (see the main text for details).

We also recall that the time dynamics of the dimer in Eq. (1) is anti-$\mathcal{PT}$ symmetric, whereas the evolution matrix $\hat{H}_{1D}$ is $\mathcal{PT}$-symmetric. That is, the exact anti-$\mathcal{PT}$ phase of the field moments corresponds to a broken $PT$-phases of $\hat{H}_{1D}$, and vice versa. This interchange between phases is caused by the imaginary factor $i$, appearing in the equation of motion.

The procedure to pass from the physics of the NHH $\hat{H}_{1D}$ to that of the field moments is the following. The intensity of a wave function $|\Psi\rangle(t)$ at a site $j$ and evolving with $\hat{H}_{1D}$ corresponds to the moment $\left\langle \hat{a}_1^j \hat{a}_2^{N-j} \right\rangle (t)$ and evolving with the Lindlbad master equation. Therefore, by measuring a given $N$th-order field moment, we can simulate the physics of the corresponding $j$th site of a $n = N + 1$ site NHH. To fix the initial condition, it is sufficient to initialize the system in the experimentally readily-accessible coherent states $|\alpha_1, \alpha_2\rangle$ (where, with this notation, we mean that the first cavity is in the coherent state $|\alpha\rangle_1$ and the second cavity is in $|\alpha\rangle_2$, where $\hat{a}_j|\alpha_j\rangle = \alpha_j|\alpha_j\rangle$). Indeed, by choosing $|\alpha_1| = |\alpha_2|$, $\left\langle \hat{a}_1^j \hat{a}_2^{N-j} \right\rangle$ is constant, and therefore this is equivalent to an initial wave function that is extended. Instead, the condition $|\alpha_1| \neq |\alpha_2|$ corresponds to a certain local-

ized initial state. In both cases, for a state to become an eigenstate, one also has to appropriately choose the phase.

In Fig. 4, we investigate the physics across the (anti)-$\mathcal{PT}$ transition using the dimer moments up to ninth order, thus simulating a lattice of size $n = 10$. We focus, in particular, on the quantum simulation of the eigenvectors $|\psi_0^R\rangle$ and $|\psi_N^R\rangle$, i.e., those showing topological properties.

At first, we can observe that the eigenmodes $|\Psi_{0,N}^R\rangle$ of $\hat{H}_{1D}$ are extended in the exact $\mathcal{PT}$ (broken anti-$\mathcal{PT}$) phase $\Delta < \gamma$. To do that, we initialize the system in the extended states $|\alpha_1 = 1, \alpha_2 = \pm 1\rangle$, for $\Delta = 0$. All the moments exhibit the same amplification (decay) with the same rate $E_0$ ($E_N$) [see Figs. 4(a),(b)]. We conclude that the initial state is an eigenstate of the system. One can, thus, verify that these modes are the slowest (largest) decaying ones (other intermediately decaying modes are not shown here). In Figs. 4(c), instead, we show that the moments of an initial state with $|\alpha_1| \neq |\alpha_2|$ evolves with different rates, meaning that a localized state it is not an eigenstate of $\hat{H}_{1D}$ in this phase. This confirms that: (i) That the eigenenergies of $\hat{H}_{1D}$ in this phase are purely real [c.f. Fig. 1(e)] and (ii) that $|\Psi_{0,N}^R\rangle$ are completely delocalized.

Let us now consider the exact anti-$\mathcal{PT}$ phase (broken-$PT$ phase) for $\Delta > \gamma$. At first, let us confirm that the previous completely delocalized states are no more the system eigenstates in this phase. To do that, in Fig. 4(d,e) we consider again $|\alpha_1 = 1, \alpha_2 = \pm 1\rangle$ as initial state. Obviously, this time, each of the moments oscillate with a different period, showing that indeed those delocalized states are not the eigenstates of the system. However, if we initialize the system either in $|1, -0.268i\rangle$ or $|-0.268, i\rangle$, according to Eq. (B2) with $N = 1$, and for $\Delta/\gamma = 2$, we can see that they produce regular oscillations where all the moments have the same oscillation frequency [see Fig. 4(f)], meaning that they are eigenstates of the evolution. That is, the formation of the two topological skin modes is manifested by the exponential difference in the "intensities" of the field moments. This demonstrates: (i) that the eigenenergies of $\hat{H}_{1D}$ have become purely imaginary, (i.e., real energy for anti-$\mathcal{PT}$-symmetric moments dynamics); (ii) the eigenstates are localized.

We remark that all these moments can be observed using standard experimental techniques in, e.g., the platforms in Figs. 1(a,b). Indeed, by checking the emitted field of each cavity in an optical implementation, or the clockwise/counter-clockwise light propagating in an optical fiber coupled to a microtoroidal resonator, it is possible to infer all the expectation values of the field inside the cavity. Furthermore, by means of recombining these emitted field and applying beam-splitter operations, one can ascertain the properties of the higher-order moments of light necessary to witness the $\mathcal{PT}$-symmetry breaking and its localization properties.

Furthermore, since the considered Lindblad master equation is quadratic in Eq. (1), this implies that the field moments dynamics also determines the time evolution of the corresponding time correlation functions of the fields, according to the quantum regression theorem [82, 104]. As such, instead of measuring the time evolution of higher-order field moments, one can measure the high-order photon two-time correlation functions.

## V. DISCUSSION AND OUTLOOK

In the studied example above, we considered the synthetic field moments space of an anti-$\mathcal{PT}$-symmetric dimer. Nevertheless, our conclusions, regarding localization transition in the synthetic chain of the fields moments, remain valid even for a general non-Hermitian dimer, where such a symmetry (including $\mathcal{PT}$) does not hold, but a system possesses an EP or an exceptional line (surface). In that case, however, the localization phase diagram can become more involved (see Appendix D).

Compared to some of the previous studies on localization transitions in the 1D non-interacting non-Hermitian systems [36, 50], whose Hamiltonians are presented via Toeplitz matrices, here, instead, we demonstrate how the emerging NHHs, defined in synthetic field moments

space of a bosonic dimer and describing a 1D model with open boundary condition, can acquire a Sylvester matrix shape. This also allows to analytically determine the eigendecomposition of the NHHs [92].

Note that the revealed above localization and topological transition can also occur in the synthetic space spanned by Hermitian field moments, e.g., moments of photon numbers, which might be easier to detect than non-Hermitian ones (though the latter can be detected via the two-time correlation functions). Moreover, this localization phenomena can be observed in synthetic space of nonlinear quadratic systems (corresponding to quantum nonlinear parametric processes), where Hermitian and non-Hermitian field moments are, in general, mixed in the system dynamics, which allows to have complex entries for the corresponding evolution matrices, which leads to the existsence of high-degenerate EPs in the system. Moreover, in dissipative and linear systems, if there is a gain in a system, there will be noise presented in the dynamics of the Hermitian moments. Nonetheless, the overall effect would be a certain nonzero shift in the moments in the long time limit and which leaves the very time dynamics unchanged [105].

Importantly, our results can be easily extended to higher-dimensional chains. For instance, instead of a dimer, one can study synthetic space of triangular or tetrahedron chains [see Fig. 5]. If these three- or four-coupled open cavities exhibit high-degenerate exceptional points, lines, or surfaces, then the resulting synthetic lattices defined by the corresponding field moments are characterized by high degeneracies, and, hence, by various localization and topological phenomena in the resulting synthetic lattices.

Apart from quadratic systems, it would be also interesting to investigate the synthetic space of moments arising in nonlinear systems, e.g., a laser or Kerr resonators. In such a system, the nonlinearity would couple moments with different orders, inducing an "extra" dimension in the synthetic space of the field moments. In this sense, it may be possible to re-interpret known spectral and continuous dissipative phase transitions where nonlinearity plays a central role [15, 106–109] and reveal, if any, topological and localization features emerging in synthetic space of such nonlinear systems.

Another interesting direction of future research will be the investigation of higher-order moments spaces of quadratic fermionic fields, and their possible similar mapping to higher-dimensional lattice systems. A foreseeable challenge towards this extension is the mathematical construction of the fermionic higher-order moments space and the associated mapping to the single-particle NHHs. Indeed, the Kronecker sum algebra, that we employed in the bosonic case, cannot be directly applied to fermionic particles.

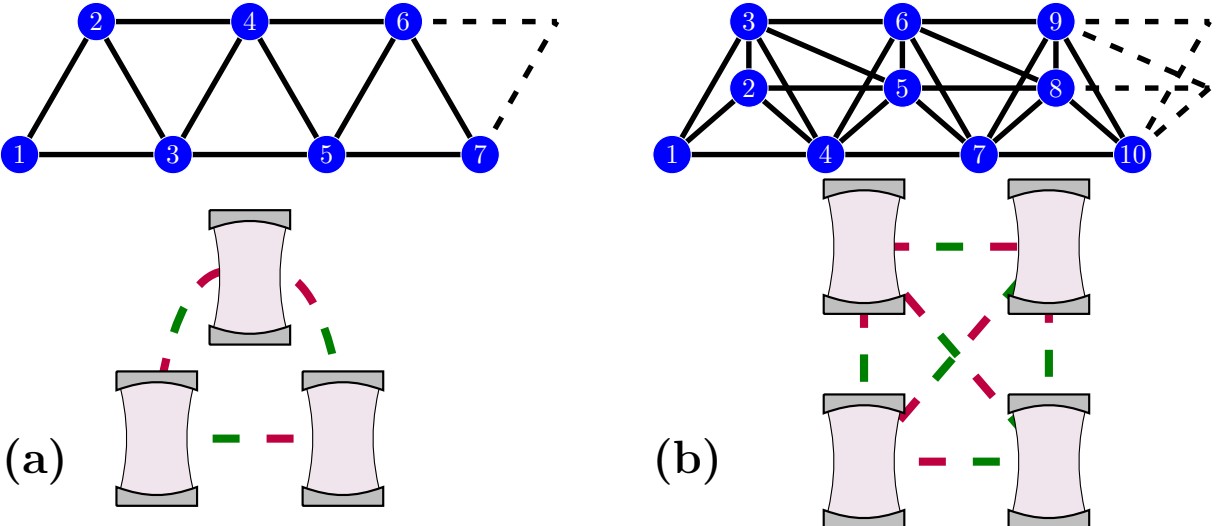

FIG. 5. Schematic representation of the emergent (a) triangular and (b) tetrahedron synthetic chains, which can arise in the higher-order field moments space of the three- and four-coupled cavities, respectively (shown in the same panels below). The site indexes in panels (a) and (b) correspond to certain (non-) Hermitian field moments of cavities, similar to that in Fig. 1.

## VI. CONCLUSIONS

In this work, we have shown that the non-Hermitian localization phenomena, such as the non-Hermitian skin effect, naturally emerge in the synthetic space of the field moments of bosonic zero-dimensional systems. As a specific example, we considered a physically realizable bosonic anti-$\mathcal{PT}$-symmetric dimer. Namely, by taking into consideration the full quantum properties of the system, we have demonstrated the emergence of spectral, topological, and localization properties in the vector space spanned by its field moments, which can be mapped onto a spatial non-Hermitian 1D chain. This synthetic field moment space exhibits a nontrivial localization transition induced by the presence of highly-degenerate EP. We revealed the emergence of $\mathbb{Z}_2$ non-Hermitian skin effect in such synthetic 1D space of high-order field moments of the dimer, i.e., the formation of skin modes appearing at both edges of synthetic $\mathcal{PT}$-symmetric 1D lattice. In comparison to previous $\mathcal{PT}$-symmetric models exhibiting localization transitions, which were proposed *a priori* and require a construction of spatial arrays of finely tuned waveguides/cavities, our method bears an opposite *a posteriori* approach, demonstrating that those theoretical models can naturally emerge in synthetic space of currently realizable low-dimensional physical systems.

Beyond the theoretical interest of this approach, it also bears a significant experimental advantage because local site perturbations can now be easily controlled. Indeed, this localization transition and $\mathcal{PT}$ symmetry breaking occurs in a 1D $N$-site synthetic space, but the physical system displaying such a property remains, in our example, a zero-dimensional bosonic dimer. Thus, to obtain a NHH akin to those emerging in higher-dimensional lattice architectures [36, 110, 111] one has to *tune the few*

*parameters of the dimer instead of fine-tuning all the parameters of the lattice.* Furthermore, as discussed in the main text, measuring the dynamics in the synthetic space just requires standard beam-splitter operations.

Our findings, thus, open avenues for further theoretical and experimental exploration of operator moments space of other (zero- and/or higher-dimensional) open quantum systems which may reveal novel topological, spectral, and localization effects. We note that our approach can be simply extended to other (low-) dimensional physical systems. For instance, the operator moments space of open three- or four-coupled cavities form synthetic non-Hermitian triangular or tetrahedron chains, respectively. Our approach is also applicable to open *nonlinear* quantum systems, where nonlinear coupling can form an extra dimension in synthetic space of operator moments of a system.

### ACKNOWLEDGMENTS

I.A. acknowledges funding by the Ministry of Education, Youth and Sports of the Czech Republic Project no. CZ.02.1.01/0.0/0.0/16_019/0000754.

### Appendix A: Mapping the space of higher-order field moments of a quadratic dimer to the one-particle 1D chain

As was shown in Ref. [73], the evolution matrix $M$, arising from the master equation in Eq. (1), and which governs a $(N + 1)$-dimensional vector of the $N$th order moments $\boldsymbol{A} = \left( \langle \hat{a}_1^N \hat{a}_2^0 \rangle, \langle \hat{a}_1^{N-1} \hat{a}_2 \rangle, \ldots, \langle \hat{a}_1^0 \hat{a}_2^N \rangle \right)^T$ via the equation of motion $i \frac{\mathrm{d}}{\mathrm{d}t} \boldsymbol{A} = iM\boldsymbol{A}$, has a tridiagonal

Sylvester matrix form, whose matrix elements read

$$M_{j,k} = -j\gamma\delta_{j,k-1} - [i(N-2j)\Delta - N\Gamma]\,\delta_{j,k}$$
$$-(N-j)\gamma\delta_{j,k+1}, \tag{A1}$$

assuming that the elements of the decoherence matrix $\hat{\gamma}$ in Eq. (1) are: $\gamma_{11} = \gamma_{22} = \Gamma$ , and $\gamma_{12} = \gamma_{21} = \gamma$. The matrix $M$ is $\mathcal{PT}$-symmetric. Indeed, by introducing the parity operator, whose matrix elements are $\mathcal{P}_{jk} = \delta_{j,N+1-k}$, and the action of the time operator $\mathcal{T}$ is complex conjugation, then it is straightforward to check that $\mathcal{PT}M\,(\mathcal{PT})^{-1} = M$. The $\mathcal{PT}$-symmetry of the matrix $M$ then automatically implies the anti-$\mathcal{PT}$-symmetry of the corresponding equation of motion of the vector of moments $\boldsymbol{A}$.

Apart from the $\mathcal{PT}$-symmetry, as has been highlighted in the main text, the matrix $M$ also has a chiral symmetry, since its eigenspectrum is symmetric, according to Eq. (3). When $M$ is a $2k \times 2k$ matrix, the chiral operator $\chi$ takes a simple matrix form with components satisfying

$$\chi_{pq} = (-1)^{p+1}i\delta_{p,2k-q-1}, \quad \text{where} \quad p,q = 0,\ldots,2k-1. \tag{A2}$$

The matrix $M$, in Eq. (A1) can be mapped onto a new 1D NHH, describing a tight-binding model of a quantum one-particle in a 1D chain with $(N+1)$ sites and with imposed open boundary conditions. To do so, one introduces a $(N+1)$-dimensional vector of annihilation operators $\hat{C} = [\hat{c}_0,\ldots,\hat{c}_N]^T$ where $\hat{c}_j^\dagger(\hat{c}_j)$ is the creation (annihilation) operator at site $j$. This new 1D NHH can be expressed as $\hat{H}_{1D} = \hat{C}^\dagger M \hat{C}$, whose operator form coincides with that in Eq. (2), assuming that $\Gamma = 0$ (as was explained in the main text). In other words, by quantizing the c-number vector $\boldsymbol{A}$, one arrives to the quantum model of a particle moving in the 1D lattice, described by the $\mathcal{PT}$-symmetric and chiral NHH $\hat{H}_{1D}$.

## Appendix B: Right eigenvectors of the Hamiltonian $\hat{H}_{1D}$ in Eq. (2), and the matrix $M$ in Eq. (A1)

As it has been found in Ref. [92], the unnormalized components of the right eigenvectors $|\psi_k^R\rangle = [\psi_k^R(0),\ldots,\psi_k^R(N)]^T$ of the $(N+1)\times(N+1)$-dimensional tridiagonal Sylvester matrix $M$ in Eq. (A1) and, consequently, the 1D NHH in Eq. (2), have the following form:

$$\psi_k^R(i) \equiv m_{ki}! \sum_{j=0}^{m_{ki}} \binom{N-M_{ki}}{j}\binom{M_{ki}}{m_{ki}-j}(-s)^{m_{ki}-j}t^j$$
$$\times \prod_{l=0}^{m_{ki}} \frac{1}{b_l} \prod_{r=m_{ki}+1}^{i} \frac{n+1-r}{b_r}t, \tag{B1}$$

where $s = \alpha - i\Delta$, $t = \alpha + i\Delta$, $\alpha = \sqrt{\gamma^2 - \Delta^2}$, $m_{ki} = \min\{k,i\}$, $M_{ki} = \max\{k,i\}$, $b_l = -l(N+1-l)\gamma$, and $b_0 = 1$. The eigenvector $|\psi_k^R\rangle$ with a quantum number $k$ corresponds to the eigenvalue $E_k$, given in Eq. (3).

Remarkably, one thus obtains analytical solution of the eigenspectrum of such a $\mathcal{PT}$-symmetric matrix of any size, hence, avoiding the need to invoke any numerical algorithms [112, 113].

The exact-$\mathcal{PT}$-symmetry of the eigenvectors $|\psi_k^R\rangle$ implies $|\psi_k^R(i)| = |\psi_k^R(N-i)|$. And chiral symmetry requires $|\psi_k^R(i)| = |\psi_{N-k}^R(N-i)|$. Note that the chiral symmetry is broken in the whole parameter space, i.e., in general $\chi|\psi_k^R\rangle = |\psi_{N-k}^R\rangle$, except the zero-energy mode for which $\chi|\psi_{N/2}^R\rangle = |\psi_{N/2}^R\rangle$, where the explicit form of the chiral operator is given in Eq. (A2).

From Eq. (B1), one finds that the eigenmodes $|\psi_{0,N}^R\rangle$ read:

$$|\psi_{0,N}^R\rangle \equiv \frac{1}{N+1}\sum_{j=0}^N u_j^{(0,N)}\hat{a}_j^\dagger|0\rangle, \tag{B2}$$

where

$$u_j^{(0,N)} = \left(i\Delta' \pm \sqrt{1-\Delta'^2}\right)^j,$$

and $\Delta' = \Delta/\gamma$. As it was stressed in the main text and Fig. 2(a), these eigenmodes capture the localization transition most vividly. Indeed, for $\Delta' < 1$ (unbroken $\mathcal{PT}$-phase), each site is equally occupied, because $u_j^{(0,N)} = \exp(\pm i\phi j)$ lie on the complex unit circle, where $\phi = \arctan\left(\Delta'/\sqrt{1-\Delta'^2}\right)$, and therefore $|u_j^{(0,N)}|^2 = 1$. At the EP, all the eigenmodes collapse to a singular extended vector $|\psi_{EP}^R\rangle$, whose amplitude at site $j$ is proportional to $(i)^j$, according to Eq. (B2). However, in the broken $\mathcal{PT}$-phase $u_j = i\beta^j$, $\beta \in \mathbb{R}$, where $\beta > 1$ ($\beta < 1$) for the state with energy $E_0$ ($E_N$), according to Eq. (B2). In other words, the skin eigenmode $|\psi_0^R\rangle$ ($|\psi_N^R\rangle$) exponentially localizes at the right (left) boundary of the chain, due to the system chirality.

## Appendix C: Effective zero-dimensional non-Hermitian Hamiltonian $\hat{H}_{\text{eff}}$, derived from Eq. (1), and its topological relation to 1D NHH $\hat{H}_{1D}$ in Eq. (2)

From the master equation in Eq. (1), one can derive an effective two-mode NHH $\hat{H}_{\text{eff}}$, which accounts for the continuous dissipative dynamics of the system, i.e., the dynamics between two quantum jumps [104]. Namely, given the decoherence matrix considered in Eq. (1), one has [104]: $\hat{H}_{\text{eff}} = \hat{H} - i\Gamma\left(\hat{a}_1^\dagger\hat{a}_1 + \hat{a}_2^\dagger\hat{a}_2\right) - i\gamma\left(\hat{a}_1^\dagger\hat{a}_2 + \hat{a}_2^\dagger\hat{a}_1\right)$, where the Hermitian Hamiltonian $\hat{H}$ is given in Eq. (1). This effective NHH determines the evolution matrix $M \equiv -iH_{\text{eff}}$ for the first-order field moments $\langle\hat{a}_j\rangle(t)$, $j = 1,2$, where $H_{\text{eff}}$ is a matrix form of the NHH in the mode representation $\hat{H}_{\text{eff}} = (\hat{a}_1^\dagger,\hat{a}_2^\dagger)H_{\text{eff}}(\hat{a}_1,\hat{a}_2)^T$. Since any evolution matrix $M$, which governs the dynamics of higher-order field moments, is obtained from the evolution matrix for the first-

order moments by simply applying Kronecker sum operation, that means that topological and spectral properties of $\hat{H}_{1D}$, in Eq. (2), originate from the NHH $-i\hat{H}_{\text{eff}}$ [73]. The topological $\mathbb{Z}_2$ invariant $\nu$ for the anti-$\mathcal{PT}$ NHH $\hat{H}_{\text{eff}}$ (i.e., when $\Gamma$ is set to zero) is determined by the sign of its determinant [32], i.e., $\nu = \text{sgn}[\det(\hat{H}_{\text{eff}})]$. Indeed, its determinant equals $\gamma^2 - \Delta^2$, meaning that $\nu = 1$ ($\nu = -1$) when the $\hat{H}_{\text{eff}}$ is exact (broken) anti-$\mathcal{PT}$-phase. This topological invariant is, thus, inherited by the synthetic $\mathcal{PT}$-symmetric NHH $\hat{H}_{1D}$. Therefore, the value of $\nu$ reflects the formation of two skin eigenmodes in the whole broken-$\mathcal{PT}$-phase of the 1D NHH $\hat{H}_{1D}$, as it was stressed in the main text. That is, although in the broken phase there might exist $O(n)$ skin eigenmodes, only two eigenvectors, namely, $|\psi_{0,N}^{\text{R}}\rangle$, remain localized at the edges in the limit $\Delta/\gamma \to \infty$, whereas the rest modes eventually migrate into the bulk. This topological resistance of the these two eigenvectors to $\Delta$ can be traced back to the NHH $\hat{H}_{\text{eff}}$, when setting $N = 1$, in Eq. (B2), which, when unfolding into higher-dimensional moments space, constitute the synthetic 1D topological edge states $|\psi_{0,N}^{\text{R}}\rangle$.

## Appendix D: Localization transition in synthetic field moments space of a non-Hermitian dimer without $\mathcal{PT}$ symmetry

Here we shed light on the localization transition phenomena in the synthetic 1D chain spanned by the higher-order field moments of a zero-dimensional non-Hermitian dimer without $\mathcal{PT}$ symmetry but with a spectral singularity (i.e., EP, exceptional line, etc.).

First, suppose that one has a bosonic dimer described by a linear NHH in the mode representation $(\hat{a}_1, \hat{a}_2)$ as follows:

$$H = \begin{pmatrix} i\omega_1 & \kappa_1 \\ \kappa_2 & i\omega_2 \end{pmatrix}, \tag{D1}$$

where, for the sake of simplicity, the parameters $\omega_{1,2}, \kappa_{1,2}$ are assumed to be real. This NHH in Eq. (D1), for arbitrary parameters, in general, does not possess a $\mathcal{PT}$ symmetry, but can have a spectral singularity expressed via equation:

$$(\omega_1 - \omega_2)^2 - 4\kappa_1\kappa_2 = 0, \tag{D2}$$

which can describe an exceptional surface. Of course the choice of the NHH in Eq. (D1) is not general, but given the fact that most of bosonic NHH are realized via induced losses, for simplicity, we can consider the NHH given in Eq. (D1).

The corresponding evolution matrix $L$ for $N$th order non-Hermitian field moments $[\langle \hat{a}_1^N \rangle, \langle \hat{a}_1^{N-1}\hat{a}_2 \rangle, \ldots, \langle \hat{a}_2^N \rangle]$ would read [73]:

$$L_{j,k} = j\kappa_2\delta_{j,k-1} + i\left[(N-j)\omega_1 + j\omega_2\right]\delta_{j,k}$$
$$+ (N-j)\kappa_1\delta_{j,k+1}, \quad j,k = 0,\ldots,N. \tag{D3}$$

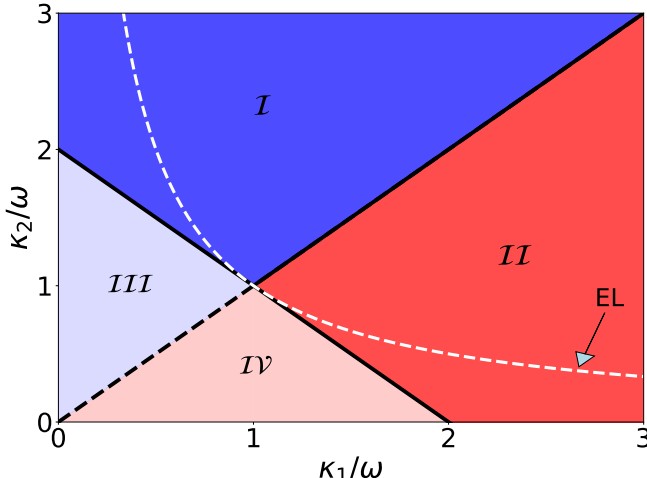

FIG. 6. Phase diagram of the right eigenmodes $\psi_n$, in Eq. (D8), comprising the eigenvectors of the synthetic chain governed by the NHH $L$ in Eq. (D3). The four different phases are separated by two lines described by equations $\kappa_1 = \kappa_2$ ($\mathcal{PT}$-symmetry condition) and that in Eq. (D11). The highly-degenerate exceptional line (EL) is shown as white dashed curve. The phases I and II are characterized by completely localized eigenmodes at the right and left edge of the chain, respectively. The phase III (IV) describes the parameter space, where the majority of eigenmodes are localized at the right (left) edge. The only mode which remains localized at the left (right) edge in the whole phase III (IV) is $\psi_N$ ($\psi_0$), and whose delocalization implies the transition border between phases III and I (IV and II).

Even though the given dimer does not possess $\mathcal{PT}$-symmetry, by appropriate transformations, one can reduce it to the $\mathcal{PT}$-symmetric one. To show that, first, and without loss of generality, assume that $\omega_1 < \omega_2$. Then by an appropriate gauge transformation $\hat{a}_{1,2} \to \hat{a}_{1,2}\exp[-(\omega_1 + \omega_2)/2t]$, one obtains:

$$H \longrightarrow H' = \begin{pmatrix} -i\omega & \kappa_1 \\ \kappa_2 & i\omega \end{pmatrix}, \tag{D4}$$

where $\omega = (\omega_2 - \omega_1)/2$. Finally, by applying a similarity transformation $S^{-1}H'S$ (taking into account that the same NHH rules the dynamics of c-number valued moments), where

$$S = \text{diag}[1, \sqrt{\kappa_2/\kappa_1}], \tag{D5}$$

one attains a $\mathcal{PT}$-symmetric NHH as follows

$$H' \longrightarrow H'' = \begin{pmatrix} -i\omega & \sqrt{\kappa_1\kappa_2} \\ \sqrt{\kappa_1\kappa_2} & i\omega \end{pmatrix}, \tag{D6}$$

with two eigenmodes $\psi_{1,2}''$ given in Eq. (B2) for $N = 1$, by identifying $\Delta = \omega$, and $\sqrt{\kappa_1\kappa_2} = \gamma$. Clearly, the eigenmodes $\psi_{1,2}''$ of the $\mathcal{PT}$-symmetric NHH $H''$ would form two topological edge modes in synthetic space of higher-order moments of the mode operators constituting the NHH $H''$, as was shown in the main text and

Appendix C. However, the original eigenmodes of the NHH $H$ in Eq. (D1) (ignoring the global dissipative rate $[\omega_1 + \omega_2]/2$) would read

$$\psi_n = S\psi_n'', \quad n = 1, 2. \tag{D7}$$

When considering the right eigenmodes $\psi_n$ of the $N$th-order ($N > 1$) moments evolution matrix $L$ in Eq. (D3), the eigenvector equations in (D7), expressed via eigenmodes of the $\mathcal{PT}$-symmetric matrix $M$ in Eq. (A1), acquire the following form:

$$\psi_n = T\psi_n'' = S_2 S_1^{-1} \psi_n'', \quad n = 0, \ldots, N, \tag{D8}$$

where $\psi_n''$ are given in Eq. (B1), and the elements of the diagonal $(N+1) \times (N+1)$ matrices $S_k$ are written as [96]:

$$S_{1,jj} = \prod_{k=1}^{j} \sqrt{\frac{M_{k,k-1}}{M_{k-1,k}}}, \quad S_{2,jj} = \prod_{k=1}^{j} \sqrt{\frac{L_{k,k-1}}{L_{k-1,k}}}, \tag{D9}$$

and $S_{k,00} = 1$, $k = 1, 2$. The elements of matrices $M_{jk}$ and $L_{jk}$ are given in Eqs. (A1) and (D3), respectively. When renormalized, the elements of the diagonal matrix $T = S_2 S_1^{-1}$, in Eq. (D8), are decreasing (increasing) from 1 to 0 (from 0 to 1) when $\kappa_1 > \kappa_2$ ($\kappa_1 < \kappa_2$), with increasing index $j$. The matrix $T$ acts as a 'squeezer' on the eigenmodes $\psi_n''$ in Eq. (D8). That is, it squeezes all the modes to the right or left edge of the chain, depending on the values $\kappa_{1,2}$.

As a result, the localization phase diagram for the right eigenmodes, comprising the synthetic 1D chain of the non-$\mathcal{PT}$-symmetric non-Hermitian system, and described by the NHH in Eq. (D3), becomes a little complex compared to its $\mathcal{PT}$-symmetric counterpart, and which is plotted in Fig. 6.

One can distinguish four different phases (see Fig. 6). The phases I and II describe completely localized modes at the right and left edge of the chain, respectively. Phases III and IV denote the parameter space where not all but a majority of the eigenmodes either tend to localized at the right and left edge, respectively. The exceptional line (EL) is shown as a white dashed curve, which separates the spectrum of the matrix $L$ in Eq. (D3), with pure imaginary-valued spectrum (parameter space below the EL) and spectrum where the real part is non-zero (parameter space above the EL).

These four phases are separated by two lines, described

by the following two equations:

$$\kappa_1 = \kappa_2, \tag{D10}$$

$$\left| \frac{\kappa_1 + \kappa_2}{2\omega} \right| = 1. \tag{D11}$$

The first equation (D10) denotes the $\mathcal{PT}$-symmetric condition. The dashed part of that line corresponds to the broken-$\mathcal{PT}$ phase, and the solid part to the exact-$\mathcal{PT}$ phase. This line divides phases I and II, as well as phases III and IV in Fig. 6.

The second line with the Eq. (D11) determines the condition when only the eigenmode $\psi_N$ ($\psi_0$), in Eq. (D8), becomes delocalized on the border of the phases I and III (II and IV), and the rest remain completely localized at the left (right) edge of the chain. Roughly speaking, that line separates phases when either of the modes $\psi_{0,N}$ abruptly change its polarization from left to right edge of the chain, or vice versa.

As can be seen from Fig. 6, in general, the EL does not separate two distinct eigenmode phases, as it is the case for $\mathcal{PT}$-symmetry. That is, when crossing the EL, the eigenmodes might not show any localization transition features (see Fig. 6).

Clearly, above the EL, the regions with phases I and II can be understood as states which are mapped from the exact-$\mathcal{PT}$ states, lying on the line $\kappa_1 = \kappa_2$, by the transformation matrix $T$ in Eq. (D8). Since in the exact-$\mathcal{PT}$ phase, all the states are symmetrical with respect to the center of the chain, their right or left localization in phases I and II, is induced by the right or left squeezing effect of the matrix $T$, respectively.

In the region below the EL, there may exist all four localized phases (see Fig. 6). In other words, the broken-$\mathcal{PT}$-symmetric states can be mapped to any phase, depending on system parameters. The latter stems from the fact that, below the EL, there is a competition between left or right localization of the broken-$\mathcal{PT}$ modes and squeezing effect of the matrix $T$, according to Eq. (D8). That is, if the squeezing is not strong enough, some of the eigenmodes will be left at the left (right) edge of the chain in the phase III (IV). And the last eigenmode which can resist that squeezing in phase III (IV) is $\psi_N$ ($\psi_0$). The region where the localization of the modes $\psi_{0,N}$ is counterbalanced with the squeezing is the line in Eq. (D11). Crossing which all eigenmodes become localized at the right or left side of the chain (phases I and II, respectively).

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
