# Peer review of "Emergent non-Hermitian skin effect in the synthetic space of (anti-)$\cal PT$-symmetric dimers"

_SciPost Physics_

## Round 3 · Referee Report · Anonymous · 2022-6-14

Strengths
1. The paper is very clearly written and the images support the text very well
2. The subject of the paper is very interesting and presents novel ideas
3. The authors propose a clear path towards realising non-Hermitian phenomena in truly quantum setups
Report
The authors introduce what they call a non-Hermitian quantum simulator, which is a Lindbladian system that can reproduce the dynamics of non-Hermitian lattices. The advantage of this method is that it requires no fine-tuning. They illustrate their method with a minimal example of an anti-PT-symmetric bosonic dimer coupled to a Markovian bath consisting of two quantum fields. They show that the synthetic space of the dimer higher-order field moments is equivalent to the non-Hermitian Hamiltonian of a one-dimensional PT-symmetric chain, and identify a localisation transition induces by the presence of an exceptional point.
This paper is very well written and very interesting. It presents a novel method to simulate non-Hermitian dynamics on a quantum level, while circumventing techniques like post-selection. I believe this paper meets all the criteria for publishing in SciPost. As such, I recommend this paper for publication.
Requested changes
1. The citation to Fig.1(d) below Eq.(3) should be Fig.1(e). Fig.1(d) should be cited elsewhere in the text.
2. I am a bit puzzled why references [110] and [111] are cited at the end of section VI. To my knowledge, they do not discuss lattices of N coupled cavities.
3. The authors focus on bosonic systems in their work. Could they comment on if and, if yes, how their work could be translated to the fermionic case?
Author: Ievgen Arkhipov on 2022-06-23 [id 2605]
(in reply to Report 2 on 2022-06-14)
We thank the Referee for his/her careful reading of our work and for the overall appreciation of the manuscript, finding it "very well written and very interesting" and recommending it for publication.
The Referee asks two minor changes and poses important and interesting questions.
We provide our answers on the Referee's comments (1-3) below.
**The referee writes:**
>The citation to Fig.1(d) below Eq.(3) should be Fig.1(e). Fig.1(d) should be cited elsewhere in the text.
**Our reply:**
We thanks the Referee for spotting that typo. We have fixed it the revised version of the manuscript.
**2. The referee writes:**
>I am a bit puzzled why references [110] and [111] are cited at the end of section VI. To my knowledge, they do not discuss lattices of N coupled cavities.
**Our reply:**
We apologize for any possible confusion. Indeed, [110] and [111] do not treat directly lattice models, but one of the motivations behind their investigation is the emergent non-Hermitian Hamiltonian describing single-particle physics of D-dimensional lattices [110] and Lieb-like lattices [111].
The revised manuscript now reads:
*Thus, to obtain a NHH akin to those emerging in higher-dimensional lattice architectures [36,110,111] one has to tune the few parameters of the dimer instead of fine-tuning all the parameters of the lattice.*
**3. The referee writes:**
>The authors focus on bosonic systems in their work. Could they comment on if and, if yes, how their work could be translated to the fermionic case?
**Our reply:**
We thank the Referee for the very interesting question. The use of higher-order moments and the mapping to the single-particle NHH is based on the bosonic operator algebra, and as such a direct extension to the fermionic problem is not possible. Nonetheless, it is an interesting question what is the structure of the moment space of quadratic fermionic systems, and it could constitute an interesting future research direction.
We added a corresponding paragraph at the end of Sec. V of the revised manuscript:
*Another interesting direction of future research will be the investigation of higher-order moments spaces of quadratic fermionic fields, and their possible similar mapping to higher-dimensional lattice systems.
A foreseeable challenge towards this extension is the mathematical construction of the fermionic higher-order moments space and the associated mapping to the single-particle NHHs.
Indeed, the Kronecker sum algebra, that we employed in the bosonic case, cannot be directly applied to fermionic particles.*
Author: Ievgen Arkhipov on 2022-06-23 [id 2606]
(in reply to Report 2 on 2022-06-14)We thank the Referee for his/her careful reading of our work and for the overall appreciation of the manuscript, finding it "very well written and very interesting" and recommending it for publication. The Referee asks two minor changes and poses important and interesting questions. We provide our answers on the Referee's comments (1-3) below.
Our reply: We thanks the Referee for spotting that typo. We have fixed it the revised version of the manuscript.
Our reply: We apologize for any possible confusion. Indeed, [110] and [111] do not treat directly lattice models, but one of the motivations behind their investigation is the emergent non-Hermitian Hamiltonian describing single-particle physics of D-dimensional lattices [110] and Lieb-like lattices [111]. The revised manuscript now reads: Thus, to obtain a NHH akin to those emerging in higher-dimensional lattice architectures [36,110,111] one has to tune the few parameters of the dimer instead of fine-tuning all the parameters of the lattice.
Our reply: We thank the Referee for the very interesting question. The use of higher-order moments and the mapping to the single-particle NHH is based on the bosonic operator algebra, and as such a direct extension to the fermionic problem is not possible. Nonetheless, it is an interesting question what is the structure of the moment space of quadratic fermionic systems, and it could constitute an interesting future research direction. We added a corresponding paragraph at the end of Sec. V of the revised manuscript: Another interesting direction of future research will be the investigation of higher-order moments spaces of quadratic fermionic fields, and their possible similar mapping to higher-dimensional lattice systems. A foreseeable challenge towards this extension is the mathematical construction of the fermionic higher-order moments space and the associated mapping to the single-particle NHHs. Indeed, the Kronecker sum algebra, that we employed in the bosonic case, cannot be directly applied to fermionic particles.

---

## Round 4 · Referee Report · Anonymous · 2022-7-1

Report
The authors have addressed my comments and questions to my satisfaction, and I believe this paper is ready for publication.

---

## Round 4 · Referee Report · Anonymous · 2022-7-25

Report
The authors show that effective non-Hermitian Hamiltonians can be simulated by using the synthetic space of field moments of a 0D bosonic system. This has several advantages, such as the possibility of obtaining effective 1D non-Hermitian systems (and their topological properties) without the need to fine-tune individual system parameters across the entire lattice. The authors demonstrate these results using a two mode system.
I enjoyed reading this paper. I think that the results are clearly presented, and the discussion is sufficiently pedagogical to make the paper accessible to nonspecialists.
After checking the citations and the previous works of the authors, however, I became confused about the novelty of these results. Upon first reading, the discussion of the abstract and introduction gave me the impression that the auhors are presenting a novel method of obtaining non-Hermitian Hamiltonians. Now, instead, my impression is that this is an incremental work, which builds on the methods and models that were already introduced by the same authors in 2102.13646 and in 2006.03557 (both published in Phys. Rev. A).
If my understanding is wrong, I would like to ask the authors to please correct me. My understanding is that the idea of simulating effective non-Hermitian systems using the space of field moments (together with its practical advantages) was already covered in these two works. Further, the same model of two incoherently coupled modes is used in all three works. The presence of nth order exceptional points is demonstrated in all three papers, and plotted as a function of the same model parameters (compare Fig. 1 of 2006.03557, Fig. 3 of 2102.13646, and Fig. 1e of this submission). I also note that in 2102.13646, it is explicitly mentioned that the authors are doing a follow-up investigation of 2006.03557. For example, the authors say:
"The model under consideration is the same as in Ref. [68]."
and
"In our previous study [68], we analyzed EPs, up to their third order, of such an anti-PT -symmetric bimodal cavity"
I did not find such statements in the current submission, giving me the mistaken initial impression that these represented novel results.
I do not believe this paper should be published in its current form. I suggest that the authors very clearly and explicitly state which results have already been obtained before (both by themselves and by others). As far as I've been able to understand from looking at all three papers, the current submission is an incremental work building on the same idea and on the same model, and therefore would not fulfill the acceptance criteria of Scipost Physics.

---

## Round 4 · Author Response

We thank the Referee for his/her careful reading of our work and for the overall appreciation of the manuscript, finding it "very well written and very interesting" and recommending it for publication. The Referee asks two minor changes and poses important and interesting questions. We provide our answers on the Referee's comments (1-3) below.
- The referee writes: The citation to Fig.1(d) below Eq.(3) should be Fig.1(e). Fig.1(d) should be cited elsewhere in the text.
Our reply: We thanks the Referee for spotting that typo. We have fixed it the revised version of the manuscript.
- The referee writes: I am a bit puzzled why references [110] and [111] are cited at the end of section VI. To my knowledge, they do not discuss lattices of N coupled cavities.
Our reply: We apologize for any possible confusion. Indeed, [110] and [111] do not treat directly lattice models, but one of the motivations behind their investigation is the emergent non-Hermitian Hamiltonian describing single-particle physics of D-dimensional lattices [110] and Lieb-like lattices [111]. The revised manuscript now reads: Thus, to obtain a NHH akin to those emerging in higher-dimensional lattice architectures [36,110,111] one has to tune the few parameters of the dimer instead of fine-tuning all the parameters of the lattice.
- The referee writes: The authors focus on bosonic systems in their work. Could they comment on if and, if yes, how their work could be translated to the fermionic case?
Our reply: We thank the Referee for the very interesting question. The use of higher-order moments and the mapping to the single-particle NHH is based on the bosonic operator algebra, and as such a direct extension to the fermionic problem is not possible. Nonetheless, it is an interesting question what is the structure of the moment space of quadratic fermionic systems, and it could constitute an interesting future research direction. We added a corresponding paragraph at the end of Sec. V of the revised manuscript: Another interesting direction of future research will be the investigation of higher-order moments spaces of quadratic fermionic fields, and their possible similar mapping to higher-dimensional lattice systems. A foreseeable challenge towards this extension is the mathematical construction of the fermionic higher-order moments space and the associated mapping to the single-particle NHHs. Indeed, the Kronecker sum algebra, that we employed in the bosonic case, cannot be directly applied to fermionic particles.

---

## Round 4 · List of Changes

1. Reference to the Fig. 1(d) has been changed to Fig. 1(e), below Eq. (3).
2. Second sentence in the second paragraph of Sec. VI has been modified.
3. A new paragraph at the end of Sec. V has been added.

You are currently on this page

---

## Editorial Decision

unknown